



# The Warm Winter Paradox in the Pliocene High Latitudes

Julia C. Tindall[1], Alan M. Haywood[1], Ulrich Salzmann[2], Aisling M. Dolan[1], and Tamara Fletcher[1]

[1]School of Earth and Environment, University of Leeds, Leeds, LS2 9JT, UK
[2]Department of Geography, Northumbria University, Newcastle upon Tyne NE1 8ST, UK,

**Correspondence:** Julia Tindall (earjcti@leeds.ac.uk)

**Abstract.** Reconciling palaeodata with model simulations of the Pliocene climate is essential for understanding a world with atmospheric $CO_2$ concentration near 400 parts per million by volume. Both models and data indicate an amplified warming of the high latitudes during the Pliocene, however terrestrial data suggests Pliocene high latitude temperatures were much higher than can be simulated by models.

Here we show that understanding the Pliocene high latitude terrestrial temperatures is particularly difficult for the coldest months, where the temperatures obtained from models and different proxies can vary by more than 20°C. We refer to this mismatch as the 'warm winter paradox'.

Analysis suggests the warm winter paradox could be due to a number of factors including: model structural uncertainty, proxy data not being strongly constrained by winter temperatures, uncertainties on data reconstruction methods and also that the Pliocene high latitude climate does not have a modern analogue. Refinements to model boundary conditions or proxy dating are unlikely to contribute significantly to the resolution of the warm winter paradox.

For the Pliocene, high latitude, terrestrial, summer temperatures, models and different proxies are in good agreement. Those factors which cause uncertainty on winter temperatures are shown to be much less important for the summer. Until some of the uncertainties on winter, high latitude, Pliocene temperatures can be reduced, we suggest a data-model comparison should focus on the summer. This is expected to give more meaningful and accurate results than a data-model comparison which focuses on the annual mean.

## 1 Introduction

Data Model Comparison (DMC) is a powerful tool in palaeoclimatology. Where data and models agree on a signal of past climate it can provide confidence in both that signal and in the accuracy of the models used for climate change research. When models and data are subject to large disagreements the opposite can occur. Unless there are well known errors or biases in

the data or models, model-data disagreement can reduce confidence in both our understanding of past climates and of model



simulations. It can also make it difficult to understand why the signals seen in the proxy record are occurring.

This paper will focus on a DMC for the Pliocene, focussing on the mid-Piacenzian warm period (mPWP, previously referred to as the mid-Pliocene warm period) which occurred between ca 3.3 - 3.0Ma (Dowsett et al., 2016), and in particular the KM5C
timeslice ($\sim$ 3.205Ma). This is the most recent example of a world which had $CO_2$ levels similar to present, and was found by Burke et al. (2018) to be the most similar geological benchmark to global surface temperature predictions of 2030 CE. It is, therefore, a crucial period for model data consensus.

The mPWP has been the subject of a co-ordinated international modelling intercomparison project, the Pliocene Model
Intercomparison Project (PlioMIP: Haywood et al., 2010). Model results from the first phase of PlioMIP (PlioMIP1) were compared with palaeodata over the ocean (Dowsett et al., 2012, 2013). It was found that the PlioMIP1 model ensemble was able to reproduce many of the spatial characteristics of SST warming, however the models could not simulate the magnitude of the warming at high latitudes.

Over land there was greater disagreement between PlioMIP1 models and data than over the oceans. Salzmann et al. (2013) assessed whether uncertainties in methodology could improve the DMC. On the data side they assessed uncertainties due to bioclimatic tolerances and dating. On the modelling side they assessed uncertainties due to orbital configuration and $CO_2$ levels. Including all of these sources of uncertainty allowed models and data to overlap in many places, however some of these uncertainties were large, meaning it would be difficult to determining the 'true' temperature. Also there were still locations
where model and data did not agree within the range of the uncertainties. At these locations Salzmann et al. (2013) noted that "the underlying reasons for these large and statistically significant DMC mismatches are unknown".

Feng et al. (2017) compared high latitude terrestrial mPWP temperature reconstructions to model simulations with the CCSM4 model. They found that the model was able to simulate the spatial patterns seen in the data, but underestimated
the magnitude of the terrestrial warming by 10°C. Sensitivity tests showed that this could be reduced by 1-2°C by changing insolation, closing Arctic gateways or by increasing $CO_2$, but model and data could not be fully reconciled.

Sensitivity studies, based on PlioMIP1 boundary conditions, have also been performed by Howell et al. (2016) and Hill (2015) in order to investigate whether changes in model forcing can improve model-data agreement. Hill (2015) found that
even after including changes to river routing, ocean bathymetry and additional land mass in the modern Barents sea, the HadCM3 model did not show improved agreement with data at the Beaver Pond site (79°N, 82°W). However he did point out that if the proxy were biased towards the summer months than model-data agreement could be possible. Howell et al. (2016) considered sensitivity to orbital forcing, atmospheric $CO_2$ and a reduced albedo of sea ice. They also found that even with the most extreme forcing the annual mean temperatures reconstructed from the proxy data at high latitudes could not be reproduced.




For the second phase of PlioMIP (PlioMIP2) substantial effort has been made to improve DMC by reducing potential sources of uncertainty attributed to a) Model boundary conditions, b) Model structure and c) data. Model uncertainties were reduced by a) utilising an improved set of model boundary conditions (PRISM4; Dowsett et al., 2016), and b) increasing the size and complexity of the PlioMIP2 ensemble relative to PlioMIP1. Although there are many sources of data uncertainty, Haywood et al. (2013) highlighted temporal uncertainty as a particular issue. PlioMIP1 focussed on a $> 200,000$ year timeslab (3.264 - 3.025Ma) within which there would be a range of climates that the data could represent, while the models would be representing a very short 'timeslice'. To improve this, PlioMIP2 model simulations represented the Marine Isotope Stage (MIS) KM5c timeslice (3.205Ma). Prescott et al. (2014) showed that the PlioMIP2 simulations could be accurately compared with data that was dated to within 20,000 years of KM5c.

Of all the changes made between PlioMIP1 and PlioMIP2, moving to the KM5c timeslice was perhaps the most controversial. Although it is desirable scientifically, it is extremely challenging to obtain proxy data to within the required temporal limits. This meant that the 100 ocean sites that were included in a DMC for PlioMIP1 (Dowsett et al., 2013), had reduced to 37 ocean sites for PlioMIP2 (Haywood et al., 2020). Over land, where the technical challenges of generating a robust age control are greater, there is inadequate data available for the KM5c timeslice with which to confront the models. Over land, it therefore remains necessary to utilise data from the mPWP, although any DMC must consider uncertainties on the age of the data.

Figure 1 shows the initial DMC for PlioMIP2 over the land and the ocean. The background colours are multi-model mean (MMM) results from PlioMIP2 (Haywood et al., 2020), while the coloured circles show the temperature anomalies obtained from proxy data at each site. Over the ocean (figure 1a), the MMM and the data are within 2°C for 23 of the 37 sites, with the MMM agreeing with the data better than any of the individual PlioMIP2 models (Haywood et al., 2020). Over land (figure 1b) model and data agree well over the Mediterranean region and southeastern Australia. However at high latitude sites the data suggests much higher temperatures than the models. The same as was found in PlioMIP1 (Salzmann et al., 2013).

Despite the limited data, figure 1b suggests that the models are unable to accurately simulate terrestrial polar amplification. If this is true it could be very concerning when simulating future climate change. It is therefore crucial to improve our understanding of why models and data do not agree at terrestrial high latitudes.

Although the ocean model-data discrepancy seen in PlioMIP1 has reduced in PlioMIP2 (Haywood et al., 2020), the terrestrial model-data discrepancy remains. In this paper we will analyse the terrestrial DMC in more detail. We will show that the model-data discrepancy is mostly confined to the high latitude winter temperatures, where temperatures from the data are greatly in excess of those from the models. This winter temperature discrepancy will be termed the 'warm winter paradox'. We will consider several possible reasons for the warm winter paradox including: model boundary condition and structural uncertainty, proxy data not being strongly constrained by winter temperatures, uncertainties on data reconstruction methods, uncertainties on proxy dating, and that in some parts of the world the Pliocene climate is outside the modern sample. We will

also show that uncertainties on summer temperatures are very different from those on winter temperatures and that a summer DMC is likely to lead to more accurate results.

The layout of the paper is as follows. Section 2 will describe the modelling and methods used. Section 3 will present a DMC focusing on both the annual mean and also on seasonal temperatures. Section 4 will discuss possible reasons for the 'warm winter paradox'. A discussion of the results and conclusions will be presented in section 5.

## 2   Methods

### 2.1   Climate Modelling

This paper makes use of two sets of modelling simulations to represent the mPWP. The first set is the model results from PlioMIP2, the second is a set of simulations run with the HadCM3 climate model to assess uncertainties caused by orbital forcing. These are described below.

#### 2.1.1   PlioMIP2 core experiments

The PlioMIP2 ensemble (Haywood et al., 2020), is the largest consistent set of mPWP model simulations to date. All modelling groups participating in PlioMIP2 were required to run a preindustrial experiment and a core mPWP experiment, which was intended to represent the KM5c timeslice (3.205 Ma). Boundary conditions for the core mPWP experiment included $CO_2$ of 400ppmv (which is within the range obtained by de la Vega et al., 2020) and a modern orbit. The land-sea mask, topography, bathymetry, vegetation, soils, lakes and land ice cover were obtained from the latest iteration of PRISM (PRISM4; Dowsett et al., 2016). It must be noted that the boundary conditions were not implemented identically in all of the PlioMIP2 models although there is substantial commonality. See papers referenced in table 1 for details of how each model implemented the boundary conditions.

#### 2.1.2   HadCM3 Orbital sensitivity experiments

The KM5c timeslice had an orbit very close to modern (Haywood et al., 2013), hence all PlioMIP2 experiments were run with a modern orbit. Over land, it is difficult to obtain data with orbital temporal precision, and in order that a DMC is even possible it is necessary to utilise data from outside the KM5c timeslice and sometimes even outside the mPWP. It is reasonable to utilise such data, provided that one is aware that this will lead to errors in the DMC. Close to the KM5c timeslice these errors are mainly due to orbital configuration, hence we include orbital uncertainties on the modelled climate when comparing with terrestrial palaeodata. Data that is further away from KM5c, such as from the early Pliocene, can also be compared with the



PlioMIP2 models, provided that one is aware of the low confidence in the results due to errors in other modelled boundary conditions (e.g. $CO_2$, ice sheets) which are difficult to quantify.

The PlioMIP2 experimental design did not include orbital sensitivity experiments. We therefore assess orbital uncertainty by
including a number of sensitivity experiments run with a single model, HadCM3 (Gordon et al., 2000). Table 2 shows the top of the atmosphere (TOA) insolation for specified times within the period 2.9Ma to 3.3Ma. The first block shows the most extreme TOA insolation for January and July at 65°N, and the second block shows the most extreme TOA insolation for January and July at 56°N. The third block shows the HadCM3 modelling sensitivity experiments that we used in this paper along with their timeslice and TOA insolation. It is seen that the orbits we use here cover relatively extreme orbits for the latitudes of interest.
The orbits representing G17 (2.950Ma), K1 (3.060Ma) and KM3 (3.155Ma) have already been discussed by Prescott et al. (2014). They all show high July TOA insolation, and K1 also shows high January TOA insolation at 65°N. Here, we use an additional orbit, 3.037Ma, which maximizes January TOA insolation at 56°N, and this orbit shows a smaller TOA insolation in July than the others used. We choose orbits which are designed to produce high TOA insolation, as these will produce warmer temperatures and would be expected to reduce the model-data disagreement seen in figure 1.


## 2.2   Vegetation modelling

We simulate mPWP vegetation, by using the PlioMIP2 climate to drive the BIOME4 mechanistic global vegetation model (Kaplan, 2001). BIOME4 has been used in many previous studies of the mPWP (e.g. Salzmann et al., 2008; Pound et al., 2014; Prescott et al., 2018), and it predicts the distribution of 28 global biomes based on the monthly means of temperature, precipi-
tation, cloudiness and absolute minimum temperature.

There are two ways to run the BIOME4 model. These are a) absolute mode or b) anomaly mode. For the absolute mode, BIOME4 is driven by direct climate model outputs for the period of interest. The anomaly mode accounts for known climate model biases that occur in the model's modern simulation and are likely to propagate through to other time periods. In anomaly
mode climate inputs to BIOME4 are obtained by calculating the modelled climate anomaly from the preindustrial and adding this onto modern observations as follows:

$$mPWP_x(input) = mPWP_x(model) - PI_x(model) + modern_x(obs) \qquad (1)$$

where $x$ is one of the BIOME4 inputs (temperature, precipitation, cloudiness, absolute minimum temperature), $input$ denotes a parameter input to BIOME4, $model$ denotes a simulated value from the multimodel mean and $obs$ is a modern dataset, which
was based on observations and created for BIOME4, as described by Kaplan et al. (2003).





In this paper we will use the anomaly mode, because this gives a more detailed representation of possible biomes, particularly at small spatial scales. However in the supplementary information we will also show results from the absolute mode to highlight that, for the mPWP, large scale features of biomes are not dependent on the methodology used.


## 3 Data-Model Comparison.

### 3.1 mPWP Mean Annual Temperature

There are 8 palaeovegetation data sites that are compared with PlioMIP2 model results in figure 1b. Figure 2 shows the DMC for each of these sites with values reported in table S1.


In figure 2 the blue symbols show the difference between the multimodel mean (MMM) PI mean annual temperature (MAT) and the modern observed MAT at the datasite. The blue dotted line shows the anomaly between the modern observations and the CRU reanalysis data and is intended to represent the error bars due to comparing modern MAT at a site to a gridbox sized area for the preindustrial. It is seen that the PI MMM MAT is in good agreement with the data suggesting that there is no

inherent model bias at these locations.

Red symbols on figure 2 show the difference between the mPWP PlioMIP2 simulations and the MAT obtained from the palaevegetation-based climate reconstruction. The red circle is the MMM and the small crosses show results from the 17 individual models. Error bars on the reconstructed temperatures due to the combined bioclimatic and temporal variability are

shown by the red dotted lines (where available).

Figure 2 shows very good model-data agreement for the mPWP at the 5 sites between 47°N and 30°S. The higher latitude sites (at 64°N, 56°N and 53°N) do not show good model-data agreement. Instead the mPWP temperature suggested by the data is substantially higher than the MMM. Although no definitive conclusions can be drawn from such a small number of

datapoints, it appears that the models have more difficulty in reproducing the data at higher latitudes than at lower latitudes. This was also shown by Salzmann et al. (2013).

### 3.2 Seasonal Temperatures

We now consider whether the mPWP model-data disagreement seen at high latitudes is uniform throughout the year or whether

it occurs preferentially in certain seasons. Figure 3 and table S2 show the PlioMIP2 model results compared with Pliocene palaeovegetation-derived temperatures for the warmest and coldest months of the year. More details about the sites used for this comparison can be found in table 3. Ideally this comparison would be for the KM5c timeslice only, however we incorporate





additional Pliocene data because only two sites can be dated close to KM5c.

For the KM5c DMC, the PlioMIP2 MMM agrees very well with the warm month temperature at Lake El'gygytgyn (data: 15-16 °C, MMM: 16.2 °C), although the warm month temperature MMM is ∼ 6°C warmer than the data at Lake Baikal (data: 15.2-17.5 °C, MMM: 22.8°C). For the cold month temperature, there is a larger discrepancy between the MMM and the data. The MMM cold month temperature is ∼ 6°C warmer than that obtained from data at Lake El'gygytgyn and ∼ 23°C colder than that obtained from data at Lake Baikal. At Lake Baikal even the warmest model (CESM2) simulated the cold month tempera-

ture ∼ 15°C too cold. The data suggests that the KM5c cold month temperature at Lake El'gygytgyn was > 30°C cooler than at Lake Baikal, however none of the models show this: all models suggest that the two sites differ in temperature by less than 6°C.

    The second block in figure 3 shows how the PlioMIP2 models compare with other Late Pliocene data. This DMC has the caveat that the modelled data represents a different temporal slice to what has been reconstructed. Because of this temporal

mismatch we would expect some model-data disagreement, however we would highlight large model-data discrepancies as problematic. For example, at Lake Baikal we have two reconstructed temperatures: one near KM5c and the other dated as 'prior to 3.5Ma' (Demske et al., 2002). Although the reconstructed temperatures at these two dates differ, this difference is relatively small compared to the large model-data discord that occurs at this site. This suggests that accounting for dating uncertainties would not be sufficient to explain the very large model-data mismatches on the cold month temperature for the

Late Pliocene Lake Baikal site.

    Further late Pliocene climatic data from Russia was obtained by Popova et al. (2012) and is compared with model results in figure 3 (sites: Mirny, Merkutlinskiy, Kabinet, Delyankir, Chernoluche, Blizkiy and 42km). These sites show similar reconstructed temperatures to those at Lake Baikal, and corroborate a strong model-data discord for the coldest month. Additional

data for the Early Pliocene (sites Tnekveem and Hydzhak) also show the same pattern, however we do note that confidence in the DMC comparison is lower for the Early Pliocene sites.

    North American sites are also included in figure 3 at Lost Chicken Mine and the Canadian Arctic sites of Meighen Island, Beaver Pond and Fyles Leaf Beds. The Canadian Arctic sites have temperatures reconstructed using two different methods:

coexistence likelihood estimation (CRACLE; Harbert and Nixon, 2015) and an open-data method based on the coexistence approach (Fletcher et al., 2017; Mosbrugger and Utescher, 1997). Regardless of the exact dating, location or reconstruction method, the DMC over North America follows the same pattern as that seen over North Asia: the models agree reasonably well with the temperature for the warm month, but modelled temperatures are too cold for the coldest month. Differences in location, proxy age or reconstruction method can affect the temperature but are not large enough to affect the general conclusion

of model-data discord.



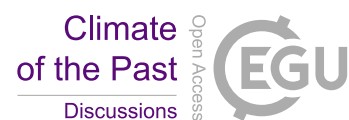

Other proxies which provide summer temperature at the Beaver Pond site agree with warm season temperatures derived from the palaeovegetation and are close to the warm month temperature from the models. These are: 1. average mean summer temperatures of 15.4 +/- 0.8°C derived from branched glycerol dialkyl glycerol tetraethers (Fletcher et al., 2019b), 2. average

growing season temperature of a) 14.2 +/- 1.3°C derived from $\delta^{18}$O values of cellulose and aragonitic freshwater molluscs and b) 10.2 +/- 1.4°C derived by applying carbonate 'clumped isotope' thermometry to mollusc shells (Csank et al., 2011). For context, the median modelled summer (JJA) temperatures at the Beaver Pond site is 10.2°C with a 20-80 percentile range of 7.8 - 14.0°C.

Other proxies which provide annual mean temperature at the Beaver Pond site agree less well with the annual mean temperatures from the models. In addition to coexistence of palaeovegetation derived temperatures (-0.4 +/- 4.1°C), Ballantyne et al. (2010) derived annual mean temperatures using oxygen isotopes and annual tree ring width (-0.5 +/- 1.9°C) and bacterial tetraether composition in paleosols (-0.6 +/- 5.0°C). These temperatures are much warmer than suggested by the models (median temperature = -11.4°C).


In addition to the above proxies, the literature contains reconstructions of both warm month and cold month Pliocene temperatures from beetle assemblages over the high latitudes of North America. However, the cold season temperatures are less well constrained than warm month temperatures (e.g. Elias et al., 1996; Elias and Matthews, 2002; Huppert and Solow, 2004). This may mean that the cold season temperature derived from beetles might reflect the modern seasonal range of temperature

in the calibration dataset rather than the Pliocene cold season (Fletcher et al., 2019a).

Figure 4 compares the PlioMIP2 models to the North American beetle assemblage data, and shows good model data agreement for the warm month temperature. Unlike the DMC for the palaeovegetation proxies, the MMM agrees reasonably well with the cold month temperature reconstructed from beetle data, particularly that derived using the Mutual Climate Range

method. However, this model-data agreement may be due to large error bars on both models and beetle data and may be due to neither of them being able to produce large enough anomalies from the modern climate (Fletcher et al., 2019a). It is unclear what causes this large disagreement on winter temperatures obtained from various sources, and we will refer to this contradiction as 'the warm winter paradox'. Resolving this 'warm winter paradox' is essential if we are to bring models into line with the data and understand the true nature of the Pliocene cold season climate.


### 3.3 Biomes

Figure 5a shows the mPWP high northern latitude biome reconstruction (Salzmann et al., 2008). This can be compared with figure 5c, which shows the biomes simulated at these locations using the MMM mPWP climate and BIOME4. The modelled biome agrees with the reconstructed biome at 14 of the 30 sites. Sites where the model suggests a different biome to that

reconstructed are shown in figure 5d. For most of the sites where the modelled biome is different to the reconstruction (par-

ticularly over North America), the reconstructed biome can be modelled at a nearby location (see figure 5b), suggesting that some discrepancies are due to small spatial errors. Over Western Europe, the warm mixed forest in the model extends too far to the east and the MMM does not reproduce the extent of the cool mixed forests seen in the data. However it is quite easy to simulate cool mixed forest in this region with only minor parameter changes to the BIOME4 model (not shown), suggesting

model and data are 'close' in this region.

A notable region of data-model mismatch is in central Eurasia. Here, the reconstructed biome is 'temperate conifer forest' and the model simulates 'evergreen taiga'. BIOME4 can only simulate 'temperate conifer forest' when the cold month temperature is above -2°C, a condition that is not provided by any of the PlioMIP2 models. The biome data-model mismatch in this

region is not easily resolved and is due to the warm winter paradox (i.e. data suggesting warmer winters than can be modelled). However, in North America and Western Europe the warm winter paradox does not prevent BIOME4 from simulating the correct vegetation biome.

## 4   What causes the warm winter paradox?

### 4.1   Could proxy dating uncertainties help resolve the warm winter paradox?

Haywood et al. (2013) suggested that the mismatch between models and data for PlioMIP1 might be caused by a comparison between model results representing a short timeslice and data that represented the ∼300,000 years of the mPWP. Moving to the KM5c timeslice for both models and data in PlioMIP2 has addressed this methodological error and there is an improvement in model data agreement for ocean proxies (Haywood et al., 2020).


A terrestrial DMC for the KM5c timeslice is problematic because there is very little terrestrial data with suitable temporal precision, and it was necessary to incorporate some data from outside the timeslice. It is therefore important to check whether the warm winter paradox could be reduced (or even eliminated) by accounting for temporal model-data mismatches.

Proxy dating uncertainties have previously been explored with modelling sensitivity experiments using different orbital configurations. These were found to show better data-model agreement in the annual mean temperature at high latitudes (e.g. Feng et al., 2017; Hill, 2015; Howell et al., 2016). It is relatively easy to increase the annual mean temperature at high latitudes by changing the orbital configuration (e.g Prescott et al., 2014), and tempting to use this as a partial solution as to why models and data do not agree. However, since the model-data mismatch occurs in the winter season, any orbital solution must increase

the cold month temperature and have a smaller effect on the warm month temperature.

Here we use the HadCM3 model to assess how different orbital configurations in the mPWP would change the warm month and cold month temperatures. The orbital configurations we include are shown in table 2. Although the list is not exhaustive, it

includes enough of the extreme orbital configurations to allow an assessment of whether orbit is likely to prove important for
resolving the warm winter paradox.

Figure 6a shows the difference between the model and data for the cold month temperature, for sites dated as 'KM5c' or
'Late Pliocene'. The HadCM3 simulation representing the KM5c timeslice is shown by the orange square, while the triangles
show the HadCM3 simulations for other timeslices considered. For context, the red circles show the KM5c simulation for other
PlioMIP2 models. As expected, the simulation, which had the largest January insolation (3.053Ma) produced the warmest cold
month temperatures. However, the cold month temperature is more sensitive to which model is used than the exact orbital
configuration. This suggests that structural model uncertainties are a more likely contributor to the warm winter paradox than
uncertainties on the exact timeslice that is to be compared with the data.

Figure 6b shows that the orbital configuration chosen can strongly affect the warm month temperature. This is unsurprising
because the summer insolation is much more variable than the winter insolation (table 2). If we had a 'warm summer para-
dox', then dating errors could be an important part of the solution. Figure 6 highlights the major shortcoming of using 'warm'
orbital configurations to improve model-data agreement for the annual mean temperature. In the annual mean both the K1 and
KM3 simulations predict higher temperatures than KM5c, and show the best agreement with the annual mean temperature
reconstructions. However neither K1 or KM3 produce a good representation of cold month or warm month temperatures. In
addition, neither of these simulations are able to simulate realistic Pliocene biomes (Prescott et al., 2018). This highlights that
a DMC on annual mean temperatures is insufficient for determining the extent of model data agreement.

This subsection asked: "Could proxy dating uncertainties help resolve the warm winter paradox?" If we assume that dating
uncertainties can be quantified by assessing the most extreme orbital configurations in the mPWP, then the answer is that proxy
dating uncertainties are unimportant for the winter season. However, the orbital configuration is not the only model boundary
condition that would change as we progress through all the timeslices that make up the Pliocene. Other boundary conditions
would include changes in trace gas, ice sheet extent, vegetation distribution, ocean gateways and associated feedbacks. Sensi-
tivity tests using different values of $CO_2$ (not shown) suggest that changing $CO_2$ would not lead to preferential warming in a
particular season. It remains to be explored whether changing other modelled boundary conditions (e.g. ice sheets) could have
a preferential effect on warming the winter season. However, the PlioMIP2 simulations only include a small ice sheet over
Greenland, hence there is limited scope for reducing ice sheets further in the Northern Hemisphere.

## 4.2 Could local climate effects help explain the warm winter paradox? (A case study of Lake Baikal)

Figure 3 shows very different results for the two sites dated near KM5c, with the PlioMIP2 models better simulating the tem-
perature at Lake El'gygytgyn than at Lake Baikal. Here we consider the DMC at Lake Baikal in more detail to assess why this





might be the case.

It is known that large bodies of water retain heat longer than the land; hence the climate around Lake Baikal is much milder
than the rest of southern Siberia. However, most models do not accurately simulate the climate stabilising effects of the lake
and their prediction of climate at this location is more representative of the wider region than the local site.

Meteorological observations for three sites near Lake Baikal are shown in table 4. Nizhneangarsk is on the northern edge
of Lake Baikal while Zhigalovo is 4° to the west and Kalakan is 7° to the east. Even though Nizhneangarsk lies between the
other two sites, the large heat capacity of Lake Baikal means that it has warmer annual mean temperature, warmer January
temperature and colder July temperature. To quantitatively estimate how much the lake will stabilise the temperature we com-
pare observations at Nizhneangarsk with the temperature interpolated onto the Nizhneangarsk location from observations at
Zhigalovo and Kalakan (see table 4). Comparing this interpolated temperature with that recorded suggests that the presence
of the lake increases the annual mean temperature by 1.8°C, it increases the January temperature by 7.8°C and cools the July
temperature by 2°C. Assuming that Lake Baikal affected the mPWP climate in an analogous way, the model results can be
corrected by this amount. This correction reduces the mPWP annual mean data-model discrepancy at this site from 8.5°C to
6.9°C, the warm month temperature data-model discrepancy from 6°C to 4°C and the cold month data-model discrepancy
from 23°C to 15°C. This correction is not sufficient to allow model-data agreement for the Pliocene winter. However it does
improve model-data agreement and will be one of a number of factors that need consideration on the Pliocene DMC.


A small caveat to this approach is that some models already account for the climate stabilising effects of the lake. For exam-
ple, CESM2 and GISS2.1G contain a lake component and both include realistic representation of lakes in their preindustrial
and their mPWP simulation. These models do not need correcting to account for the climate stabilising effects of the lake, and
applying such a correction would reduce agreement with observations for the modern. Ultimately it is a choice for individual
modelling groups as to whether their model output needs correcting to account for microclimate effects at a specific location.
In our study we suggest that the MMM temperatures at Lake Baikal requires a 'lake' correction because it is required for the
majority of the PlioMIP2 models.

### 4.3   Can other uncertainties on reconstructed temperatures help explain the warm winter paradox?

#### 4.3.1   Vegetation proxies may not be strongly related to the cold month temperature

The cold month temperatures from the PlioMIP2 models are lower than reconstructed from data. Over north Asia this leads
to a mismatch between reconstructed biomes and biomes simulated by BIOME4. However, both the cold month temperature
reconstructions and the BIOME4 model assume that the cold month temperature is a strong constraint on the distribution of
tree species, and that the limits on the range of trees can be determined using correlations from the modern climate. In fact





these assumptions may not hold. A case study from Korner et al. (2016) found that for temperate tree species, low-temperature extremes in winter (when the species were dormant) have very little relationship to range limits and that tree species could tolerate much cooler winter temperatures than those that are currently experienced. Spring temperature was found to be far more important for determining whether a species can survive and reproduce, and growing season temperature is also important. This suggests that the uncertainties on winter temperatures may be much larger than is sometimes reported.


### 4.3.2    Possible errors on reconstruction methods

Palaeoclimate reconstruction methods can be used to reconstruct modern climate. These modern reconstructions can be compared to modern observations to provide error bars on the method. Following this approach, Harbert and Nixon (2015) found the average error on the MAT reconstructed using the CRACLE method[1] was 1.3 - 1.4°C, which compared well with errors of
1.8°C for CLAMP and 2°C for leaf margin analysis. None of these errors are large enough to notably contribute to the data-model mismatch found for the Pliocene. However these errors are global averages, and do not appear uniformly over the globe. (Harbert and Baryiames, 2020, ; their figure 2) shows that the error in reconstructing the minimum temperature of the coldest month appears larger at cold temperatures than the average error over the globe. For example, sites with minimum temperature below -20°C appear to have a clear warm bias, which also occurs on the mean annual temperature. No clear bias is apparent
when reconstructing the maximum temperature of the warmest month. If this warm bias on the minimum temperature is robust, and also occurs in the Pliocene temperature reconstruction, it could contribute to the model-data discord seen in figures 2 and 3.

### 4.3.3    Different proxies suggest different cold month temperatures

We note from figures 3 and 4 that there are differences in the temperature reconstructions from different proxies. It is beyond the
scope of this paper to compare and contrast proxy methods and this subject is covered in other papers (e.g. Harbert and Nixon, 2015; Fletcher et al., 2019a). However two things are of note. Firstly, the only two sites that are close to KM5c on figure 3, Lake Baikal and Lake El'gygytgyn, have similar warm month temperature reconstruction, but suggest cold month temperatures that differ by over 30°C, a feature that does not occur in any of the models or in modern observational data. Could some of this difference between the two sites be related to the different methodologies used for temperature reconstruction (table 3)?
Secondly, differences between proxy reconstructed temperature for the Pliocene are often larger than published error bars (or may result from some published ranges not including error bars). For example, the cold month mean temperature from the coexistence likelihood estimation provides a warmer temperature than the coexistence approach, which provides a warmer temperature than the Mutual Climatic Range method for beetle assemblages (figures 3 and 4). For the warm month mean temperature all approaches yield similar temperatures. Note that here we are not suggesting that one reconstruction of the cold
month mean temperature is better than another, instead we are pointing out that the cold month temperature from proxy data

---

[1]this is labelled 'Coexistence Likelihood Estimation in figure 3, and a similar method (see Klages et al., 2020) was also used at Lake Baikal for KM5c





appears to be subject to greater uncertainty than the warm month temperatures.

## 4.4   Could modelling errors be responsible for the warm winter paradox?

Models are, by their nature, an imperfect representation of reality and all models have errors, even for the preindustrial where
boundary conditions are well known and where some model parameters have been chosen based on their ability to produce
a realistic climate. Pliocene simulations use the same model parameters that have been optimised for the modern climate and
also have less well constrained boundary conditions. Hence, the simulated Pliocene climate contains more uncertainties than
the corresponding preindustrial simulations.

Figure 3 shows that across the PlioMIP2 ensemble there is large variation in the simulated CMMT temperatures (up to $\sim$
$20°C$). This large range is from a suite of models that have been run with very similar boundary conditions (orbit, $CO_2$, ice
sheets), so the model spread is likely due to inherent model structure. The PlioMIP2 models have equilibrium climate sensitiv-
ities[2], between $2.3°C$ and $5.2°C$ which covers the range suggested by IPCC, hence the ensembles response to $CO_2$ forcing is
likely reasonable. However, the modelled response to the full Pliocene boundary condition changes (e.g. ice sheets and orog-
raphy) is less constrained by other sources. There may also some important forcings (e.g. Methane; Hopcroft et al., 2020) that
have not been included in the PlioMIP2 simulations, and some important feedbacks, for example fire (Fletcher et al., 2019b)
and chemistry (Unger and Yue, 2014), that are not included.

Clouds and convection feedbacks are subject to uncertainties and could lead to errors in Pliocene simulations. Abbot and Tziperman
(2008) used a single column model to show that deep atmospheric convection might occur during winter in ice-free high lati-
tude oceans, and could increase high latitude winter temperatures by $\sim 50°C$. However, this feedback did not occur in any of
the PlioMIP2 models despite January Arctic sea ice extent being reduced by up to 76%.

Another potential source of model error might be that the PlioMIP2 models are not high enough resolution to fully resolve
the processes occuring in the Pliocene. For example, Arnold et al. (2014) showed that modelling a high $CO_2$ world with a
cloud-permitting model led to greater Arctic cloud cover and sea ice loss than if convection were parameterized. However,
these changes had relatively minor effects on Arctic temperatures.

Pope et al. (2011) considered the uncertainty that could result from incorrect tuning of the model parameters in the HadCM3
model by running a Pliocene perturbed physics ensemble which varied uncertain model parameters (within reasonable bounds).
They showed that the effect of using model parameters designed to produce a high sensitivity climate could be substantial (ap-
proximately 2-5°C of warming in the Pliocene over the high latitude continents). However they presented their results for the
annual mean temperature only, so we currently do not know how this increase would manifest seasonally. They also noted that

---

[2]Equilibrium Climate Sensitivity is defined as the global temperature response to a doubling of $CO_2$ once the energy balance has reached equilibrium





if this 'high sensitivity' climate was used to drive BIOME4 then the biome distribution did not agree with reconstructions as
well as the biome distribution simulated from the control climate.

It is likely that as models develop there will be future refinements to the Pliocene model simulations, and this could provide
a part of the solution to the warm winter paradox. However, the current set of PlioMIP2 experiments provides good agreement
with ocean data (McClymont et al., 2020; Haywood et al., 2020), so the potential for model refinements are subject to ocean
data constraints and may not change as much as is needed to fully agree with the cold month terrestrial temperature data.

## 4.5   Could a geographical shift in biome boundaries explain the warm winter paradox?

Figure 7 shows that in the Pliocene the high latitude forests were further poleward than they were in the preindustrial climate.
This is logical, because in a warmer climate vegetation would be able to inhabit regions that are too cold today. However, this
does not mean that the climate experienced by a biome in the Pliocene will be exactly the same as the climate experienced by
that biome today.

Figure 8 shows the incoming solar insolation at the top of the atmosphere for each month and latitude. For clarity this has
been normalised by the incoming solar radiation at 55°N. We see that in May/June/July the insolation at all latitudes shown
is similar to the insolation at 55°N, while in other months (particularly the winter) the insolation (relative to that at 55°N)
decreases dramatically as we move to higher latitudes. Because KM5c has a near modern orbit, figure 8 applies to both KM5c
and the modern, and is one of the most certain features of the KM5c world.

We can be confident that if a plant species occupied a higher latitude niche at KM5c than it does today then the amount of
incoming solar insolation it experienced would vary more throughout the year. As an example, Fletcher et al. (2019b) showed
that the Pliocene floral assemblage at Beaver Pond ($\sim 79°N$) most closely resembles modern vegetation found in northern
North America particularly on the Eastern Margin, the Western Margin and Fennoscandina. All of these locations are at lati-
tudes $< 70°N$ and some are at latitudes $< 50°N$. It is seen in figure 8 that these lower latitude analogues will have a much less
extreme seasonal cycle of insolation than Beaver Pond. Of course, climate feedbacks at the Pliocene Beaver Pond site could
counteract the seasonal cycle in insolation and allow the seasonal temperature cycle to become similar to the modern climate
at a lower latitude. However, it is likely that this would not be the case for every location over the globe, and some mid-high
latitude ecosystems in the mid-Pliocene could experience environmental conditions outside the modern sample. This would
lead to uncertainties on climate reconstruction methods that utilise information from the modern distribution of plants to de-
termine past climates. Any such uncertainties would increase error bars on winter temperatures because plant distributions are
more strongly constrained by spring and summer temperatures (Korner et al., 2016). Furthermore the error bars would likely be
skewed towards colder temperatures because winter insolation becomes strongly reduced as we move to higher latitudes. We
therefore highlight the geographical shift in biome boundaries and the potential for a non analogue climate as another possible



contributor to the warm winter paradox.

## 5 Discussion and conclusions

The latest iteration of the Pliocene Modelling Intercomparison Project (PlioMIP2) produces temperatures that agree very well with proxy data over oceans the tropical land surface and the high latitude land surface WMMT. The high latitude CMMT, however, shows large model-data disagreement. The proxy data suggests very high temperatures that the models are unable to replicate. We term this the 'warm winter paradox'.

This cold month, high latitude, terrestrial data-model discord is not unique to the Pliocene. For the Holocene, Mauri et al. (2015) noted that their "climate reconstruction suggests warming in Europe during the mid-Holocene was greater in winter than in summer, an apparent paradox that is not consistent with current climate model simulations and traditional interpretations of Milankovitch theory". For the LGM, Kageyama et al. (2021) showed that none of the models analysed could simulate the amplitude of the reconstructed winter cooling over Western Europe. For older, greenhouse climates in the Mesozoic and early Cenozoic there has been a longstanding 'equable climate problem' (e.g. Greenwood and Wing, 1995; Huber and Sloan, 2001), where models typically predict temperatures 20°C colder than data over the continental interiors. Huber and Caballero (2011) showed that modelling the Eocene with very large $CO_2$ values ($16 \times$ preindustrial) was able to simulate cold month temperatures in reasonable agreement with the data. However studies of the Eocene climate generally use much smaller $CO_2$ forcing ($1 - 9 \times$ preindustrial; Lunt et al., 2021).

For the Pliocene we have investigated several possible contributors to the 'warm winter paradox'. It is likely that the 'warm winter paradox' cannot be solved by one single factor, and instead that it is due to a multitude of factors. The relevant factors we have considered, do lead to a potential warm bias on the data and a potential cold bias on the models, suggesting they could increase model-data agreement.

For the warm winter paradox, we find that structural model uncertainties are likely to be more important than uncertainties in the model boundary conditions. This is because the data-model discord does not seem to be largely dependent on the exact age of the proxy data, or simulated orbital boundary conditions yet the range of temperatures simulated by different models is relatively large. All models also share some aspects of structural uncertainty that could affect the simulated climate. For example none are able to fully resolve convection or other high resolution processes.

From a data perspective we have noted that different data sources provide very different reconstructions of winter temperatures. Although, there are good reason to suggest that some reconstructions are better than others, the very different reconstructed temperatures lend some uncertainty to this winter temperatures. Additional uncertainty arises because, the proxies we





have considered, (vegetation and beetle assemblage) may not be particularly sensitive to cold month temperatures.

The methodology of obtaining temperatures may contribute errors to the DMC. For example, a modern day test case showed that the CRACLE method had a warm bias on CMMT of 4.4°C for very cold winter temperatures, this was more than 3 times

that global average error. In the models, very local effects that the models do not resolve could bias results, as was evidenced by a modelled CMMT cold bias of 7.8°C at the Lake Baikal site. Removing these two potential methodological errors would bring the model and data 12.2°C closer together at Lake Baikal.

Finally we considered the non-analogue nature of the Pliocene climate and how this might influence the DMC. If this were

an issue it would affect temperatures reconstructed from data, because temperature reconstruction methods rely on modern habitats of flora and fauna to determine range limits, which can then be used to determine Pliocene climate. However, there are likely to be Pliocene climates that are outside the modern range. At such places, the reconstructed temperatures will be subject to greater uncertainty. We have argued in this paper that the increase in uncertainty is likely to take the form of a warm (rather than a cold) bias and could provide a nudge towards greater model-data agreement.


Relative to the cold month temperature, there appear to be fewer uncertainties on the warm month temperature. Previous studies (e.g. Abbot and Tziperman, 2008) do not note as large a sensitivity on the warm month temperature to the changing climate. Proxies are more sensitive to the warm month temperature and can therefore be used to produce a more accurate reconstruction. In contrast to the cold month temperature, boundary conditions do appear important for simulating the warm

month temperature, suggesting that modelling sensitivity experiments could be used to fine tune warm month temperature and produce good model-data agreement. However, this potential to easily bring model and data into line for the warm month temperature is not needed. The PlioMIP2 models agree well with the warm month temperature from the data, and data from different sources concur.

The high latitude mPWP CMMT obtained from models and from data are so different that they cannot both be correct. Indeed, given the large uncertainties on both model and data, it is plausible that the mean value obtained from both methods are wrong, although it is not yet possible to state how large the errors on either model and data are likely to be. Until this uncertainty is reduced it might be advisable to discuss mPWP high latitude climate in terms of more consistent parameters such as WMMT or vegetation biomes. This is not to say that winter temperatures should be ignored. However we want to avoid

suggestions that one may take from such comparisons: for example that models cannot accurately simulate polar amplification. A more accurate conclusion would be that, for the Pliocene, models are very good at simulating polar amplification for the summer months, and the uncertainties from both models and data on winter temperatures are currently too large to be able to provide reliable conclusions.



*Author contributions.* JCT analysed the PlioMIP2 model simulations to produce the DMC, with advice from all coauthors. US provided the KM5c data at Lake Baikal. JCT prepared the paper with contributions from all coauthors.

*Competing interests.* The authors declare that they have no conflict of interest

*Acknowledgements.* JCT, AMH and AMD acknowledge the FP7 Ideas programme: European Research Council (grant no. PLIO-ESS, 278636) and the Past Earth Network (EPSRC grant no. EP/M008.363/1). JCT was also supported through the Centre for Environmental Modelling and Computation (CEMAC), University of Leeds.





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

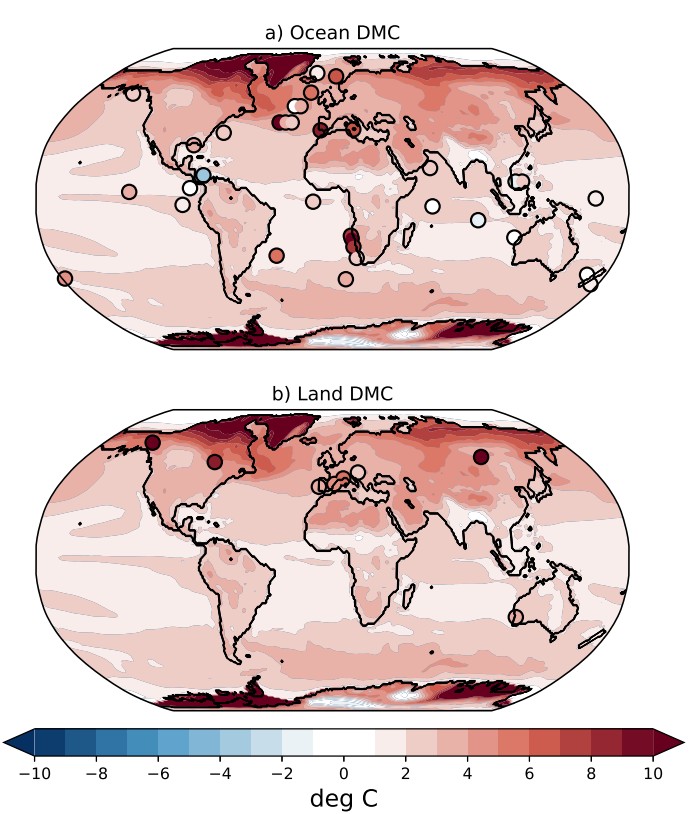

**Figure 1.** The background colours are multi-model mean results from PlioMIP2 (Haywood et al., 2020). The ocean site data SST anomaly is the difference between the McClymont et al. (2020) dataset and years 1870-1899 of the NOAA Extended Reconstructed Sea Surface Temperature (ERSST) version 5 dataset (Huang et al., 2017). The terrestrial data SAT anomaly is the difference between the KM5c terrestrial dataset and the CRU reanalysis data (CRU TS v 4.04; Harris et al., 2020) averaged over the period 1901-1930.

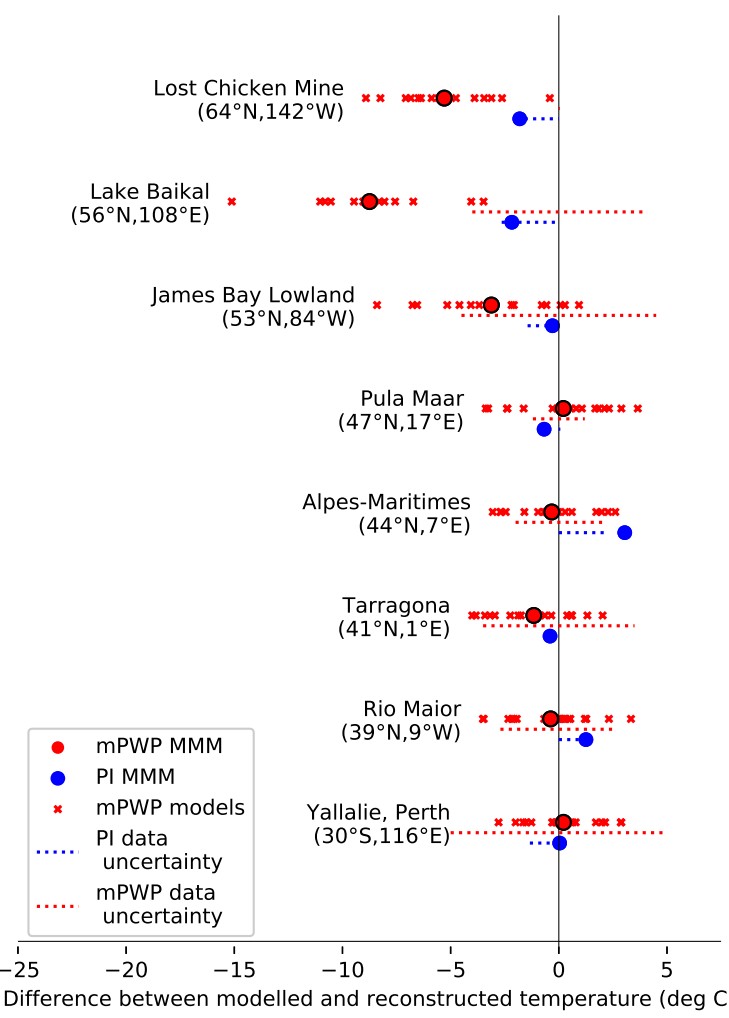

**Figure 2.** Blue circles shows the difference between MMM preindustrial MAT and modern observations at the site. The blue dotted line shows the difference between the CRU MAT for years 1901-1930 and the modern observations at the site, representing an estimate of the error caused by comparing a single modern data point to a preindustrial model gridbox. Red circle shows the difference between MMM mPWP MAT and the MAT reconstructed from the proxy; red crosses show the anomalies for the individual models. The red dotted line shows the uncertainty on the data reconstruction (where available).



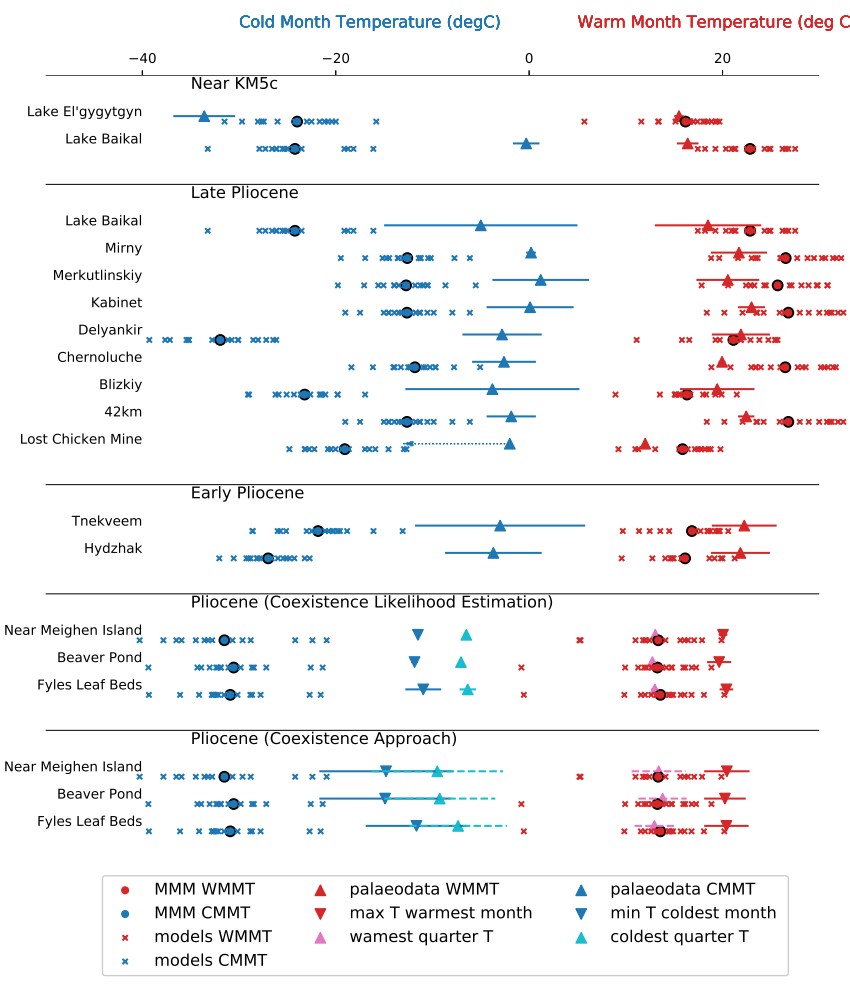

**Figure 3.** Data model comparison for the cold month mean temperature (CMMT; blue) and the warm month mean temperature (WMMT; red). Triangles show temperatures from proxy data with published uncertainties (where available). The Pliocene CMMT at Lost Chicken Mine was reported as "less than -2°C but not nearly as cold as the modern" and this is represented by the dotted line and arrow. The location of Meighen Island (80°N, 99°W) was an oceanic gridbox in most models, hence Meighen Island data has been compared to a nearby land gridbox. Model data is WMMT or CMMT and is shown by circles (MMM) or small crosses (individual models). Additional metadata is in table 3

.



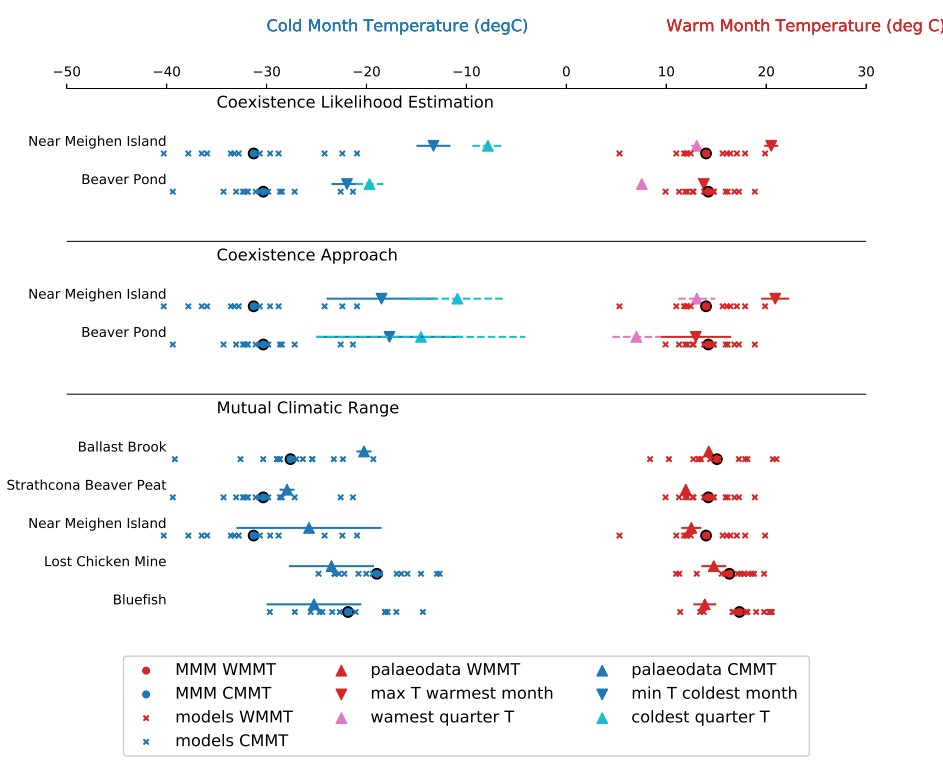

**Figure 4.** Data model comparison for the cold month mean temperature (CMMT: blue symbols) and the warm month mean temperature (WMMT: red symbols) for the Pliocene. Triangles show temperatures from beetle assemblage proxy data, obtained using various methods with published uncertainties (where available). Model data is shown by circles (MMM) or small crosses (individual models). The location of Meighen Island (80°N, 99°W) was an oceanic gridbox in most models, hence Meighen Island data has been compared to a nearby land gridbox. Additional metadata for this figure is in table 3

.





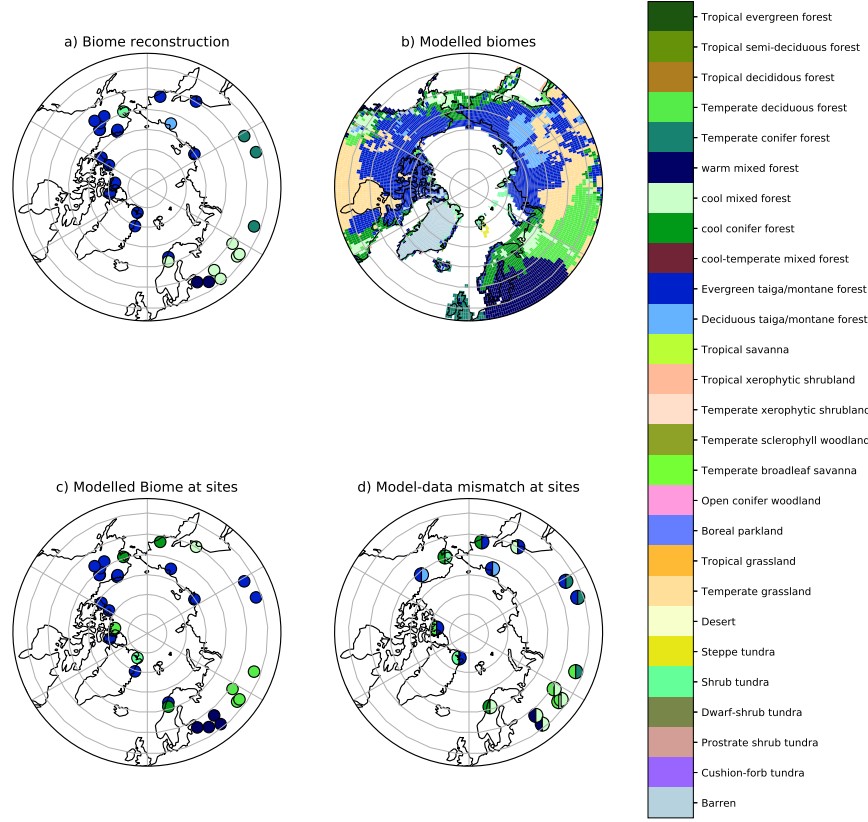

**Figure 5.** a) The reconstructed biomes from Salzmann et al. (2008). b) the modelled biomes, obtained by using the PlioMIP2 MMM to drive the BIOME4 model. c) The modelled biomes in (b), at the locations where data is available. d) locations where the reconstructed biomes do not match the modelled biome. The modelled biome is the left of the semicircle, the reconstructed biome is the right of the semicircle.



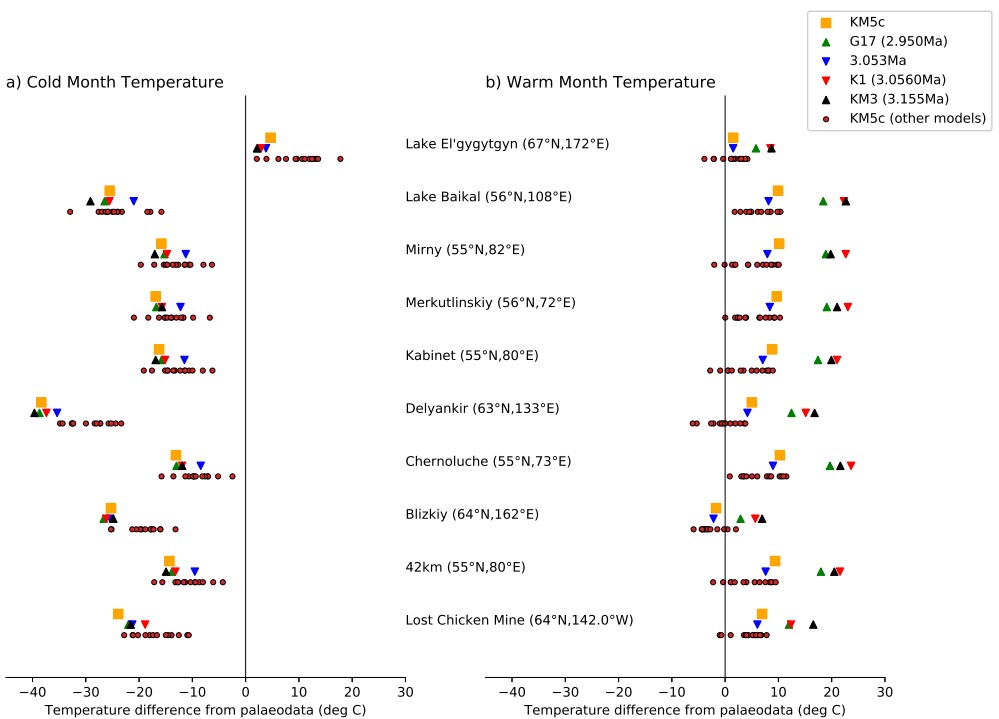

**Figure 6.** Shows the difference between the modelled temperature and the palaeodata at KM5c (Lake Baikal and Lake El'gygytgyn) and other late Pliocene sites. HadCM3 simulations with a range of different orbital configurations are shown by the square and triangles. The KM5c simulation for other models is shown by the circles.




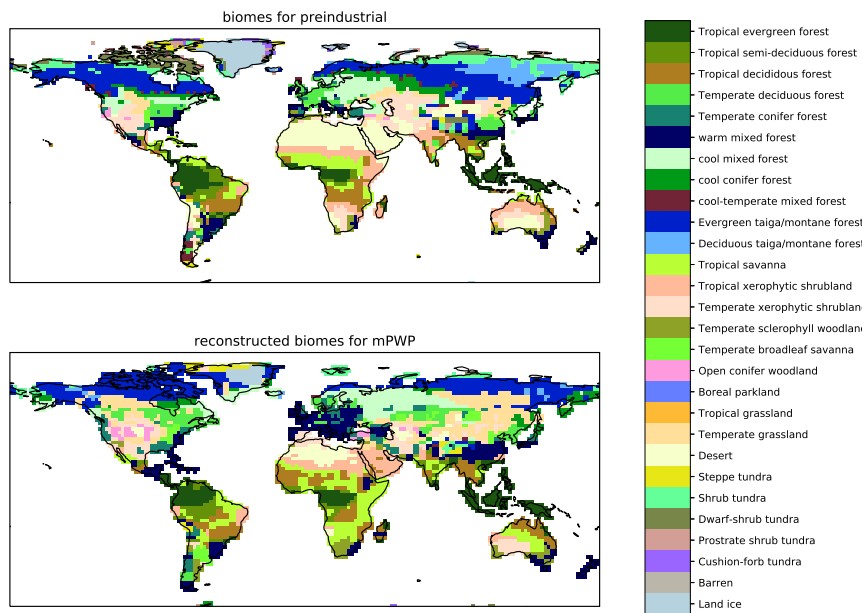

**Figure 7.** A comparison of modern biomes with the reconstructed biomes for the mPWP (Salzmann et al., 2008)

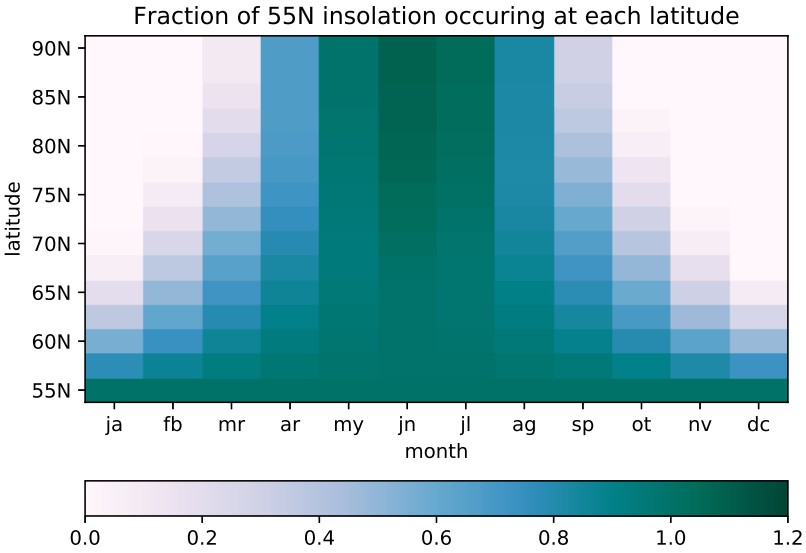

**Figure 8.** TOA insolation by latitude and month. This has been normalised by dividing the insolation for a latitude and month by the insolation for that month at 55°N

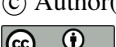



| Model name | Institution | PlioMIP2 reference |
| --- | --- | --- |
| CCSM4-NCAR | NCAR, USA | Feng et al. (2020) |
| CCSM4-Utrecht | Utrecht University, The Netherlands | Baatsen et al., (in review) |
| CCSM4-UofT | University of Toronto, Canada | Chandan and Peltier (2017) |
| CESM1.2 | NCAR, USA | Feng et al. (2020) |
| CESM2 | NCAR, USA | Feng et al. (2020) |
| COSMOS | Alfred Wegener Institute, Germany | Stepanek et al. (2020) |
| EC-Earth3.3 | Stockholm University, Sweeden | Zheng et al. (2019) |
| GISS-E2-1-G | NASA/GISS, USA | Chandler et al. (in prep) |
| HadCM3 | University of Leeds, UK | Hunter et al. (2019) |
| HadGEM3 | University of Bristol, UK | Williams et al. (2021) |
| IPSLCM5A | LCSE, France | Tan et al. (2020) |
| IPSLCM5A2.1 | LCSE, France | Tan et al. (2020) |
| IPSLCM6A | LCSE, France | n/a |
| MIROC4m | CCSR/NIES/FRCGC, Japan | Chan and Abe-Ouchi (2020) |
| MRI-CGCM2.3 | Meteorological Research Institude, Japan | Kamae et al. (2016) |
| NorESM-L | NORCE, Norway | Li et al. (2020) |
| NorESM1-F | NORCE, Norway | Li et al. (2020) |

**Table 1.** Models participating in PlioMIP2 used in this study

|  | TIME (Ma) | JAN 65°N (W/m$^2$) | JUL 65°N (W/m$^2$) | JAN 56°N (W/m$^2$) | JUL 56°N (W/m$^2$) |
|---|---|---|---|---|---|
| max / min insolation at 65N | | | | | |
| max Jan 65°N | 3.057 | **11.8** | 506 | 58 | 515 |
| min Jan 65°N | 2.953 | **3.9** | 460 | 49 | 465 |
| max Jul 65°N | 3.037 | 8.6 | **523** | 52 | 531 |
| min Jul 65°N | 3.142 | 8.4 | **437** | 59 | 444 |
| max / min insolation at 56N | | | | | |
| max Jan 56°N | 3.053 | 9.9 | 455 | **60** | 460 |
| min Jan 56°N | 2.950 | 4.3 | 477 | **48** | 484 |
| max Jul 56°N | 3.037 | 8.6 | 523 | 52 | **531** |
| min Jul 56°N | 3.059 | 10.9 | 513 | 56 | **438** |
| insolation for orbits used in this paper | | | | | |
| KM5c | 3.205 | 6.6 | 472 | 53 | 478 |
| K1 | 3.060 | 10.1 | 508 | 54 | 521 |
| G17 | 2.950 | 4.3 | 477 | 48 | 484 |
| KM3 | 3.155 | 6.2 | 499 | 49 | 509 |
| max Jan 56°N | 3.053 | 9.9 | 455 | 60 | 460 |

**Table 2.** TOA insolation at various times in the mPWP. The TOA insolation for each orbit was obtained using Laskar et al. (2004).





| SITE (LOCATION) | PROXY TYPE | REFERENCE | AGE |
|---|---|---|---|
| Lake El'gygytgyn (67°N 172°E) | pollen: BMA | Brigham-Grette et al. (2013) Pavel Tarasov (pers. comm) | 3.199-3.209Ma |
| Lake Baikal (56°N, 108°E) | vegetation CLE | unpublished (method of Klages et al. (2020) and Hyland et al. (2018)) | KM5c |
| Lake Baikal (56°N, 108°E) | vegetation CA | Demske et al. (2002) | prior to 3.5Ma |
| Mirny (55°N, 82°E) | vegetation CA | Popova et al. (2012) | Late Pliocene |
| Merkutlinskiy (56°N, 72°E) | vegetation CA | Popova et al. (2012) | Late Pliocene |
| Kabinet (55°N, 80°E) | vegetation CA | Popova et al. (2012) | Late Pliocene |
| Delyankir (63°N, 133°E) | vegetation CA | Popova et al. (2012) | Late Pliocene |
| Chernoluche (55°N, 73°E) | vegetation CA | Popova et al. (2012) | Late Pliocene |
| Blizkiy (64°N, 162°E) | vegetation CA | Popova et al. (2012) | Late Pliocene |
| 42km (55°N, 80°E) | Vegetation CA | Popova et al. (2012) | Late Pliocene |
| Lost Chicken Mine (64°N, 142°W) | vegetation QE Beetle: MCR | Ager et al. (1994) Matthews Jr. and Telka (1997) | 2.9 +/- 0.4Ma |
| Tnekveem (66°N, 177°E) | vegetation CA | Popova et al. (2012) | Late Pliocene |
| Hydzhak (63°N, 147°E) | vegetation CA | Popova et al. (2012) | Late Pliocene |
| Near Meighen Island (77.5°N, 99°W) | Vegetation CLE Vegetation CA Beetle CLE Beetle CA Beetle MCR | Fletcher et al. (2017) Fletcher et al. (2019a) Elias and Matthews (2002) | (3.2-2.9Ma or 3.4Ma) Barendregt et al (submitted) |
| Beaver Pond (79°N, 82°W) | Vegetation CLE Vegetation CA Beetle CLE Beetle CA Beetle MCR | Fletcher et al. (2017) Fletcher et al. (2019a) Matthews Jr. and Fyles (2000) | 3.9 + 1.5 / -0.5 Ma |
| Fyles Leaf Bed (79°N, 83°W) | Vegetation CLE Vegetation CA | Fletcher et al. (2017) | 3.8 + 1.0 / -0.7 Ma |
| Ballast Brook (74°N, 123°W) | Beetle: MCR | Fyles et al. (1997) | 3.5Ma |
| Bluefish (67°N 139°W) | Beetle: MCR | Matthews Jr. and Telka (1997) | Late Pliocene |

**Table 3.** Metadata for the DMC in figures 3 and 4. BMA - Best Modern Analogue (Overpeck et al., 1985), CA - Coexistence approach based on Mosbrugger and Utescher (1997), CLE - Coexistence Likelihood Estimation, QE - Qualitative estimates using modern analogues, MCR - Mutual Climatic Range



| STATION | LOCATION | annual mean | January Mean | July Mean |
|---|---|---|---|---|
| Nizhneangarsk | 55.8°N 109.6°E | - 3.6°C | -22.4°C | 15.0°C |
| Zhigalovo | 54.8°N 105.2°E | -4.5°C | -28.3°C | 17.6°C |
| Kalakan | 55.1°N 116.8°E | -8.0°C | -35.7°C | 16.4°C |
| Interpolated* | 55°N, 109.6°E | -5.4°C | -30.2°C | 17.0°C |

**Table 4.** Station temperature data near Lake Baikal averaged over 1950-1970. * Interpolated data is from Kalakan and Zhigalovo.