# Peer review of "The Warm Winter Paradox in the Pliocene Northern High Latitudes"

_Climate of the Past, 2021_

## Author Response (AR1)

Please note that line numbers refer to the marked-up version.

Response to RC1

We would like to thank the reviewer for reading our manuscript and providing such positive feedback. We are very pleased that the reviewer thought our study was a useful contribution to the warm Pliocene climate.

Response to EC1

Thank you for providing comments on our paper. We have addressed the comments as follows.

- Although CMMT and WMMT were written out in full in the figure captions, it might be a good idea to also write them out in full when they first appear in the text, eg line 395.

These abbreviations were not used very often in the paper. We have therefore removed the abbreviations and written them out in full (e.g line 405, 468).

- Line 161: concerning mismatches between the reconstructed and simulated biomes in central Eurasia, are there any clues as to what the causes could be? eg the temperature, precipitation and the nature of the biomes in that region.

We briefly discussed the cause in lines 270-272: "Here, the reconstructed biome is 'temperate conifer forest' and the model simulates 'evergreen taiga'. BIOME4 can only simulate 'temperate conifer forest' when the cold month temperature is above -2∘C, a condition that is not provided by any of the PlioMIP2 models. The biome data-model mismatch in this region is not easily resolved and is due to the warm winter paradox (i.e. data suggesting warmer winters than can be modelled)."

- Line 266: Are the correct biomes in North America and Western Europe all evergreen taiga/montane forest? In fact, it seems Eastern Europe, to the east of the Baltic Sea, also shows good agreement in the biomes.

The editor is correct that over North America (and Greenland) both model and data show large areas of Evergreen taiga/montane forest, and the model-data agreement is on this biome. Where the biome is different the model and data disagree. The sites where model and data do not agree over North America are:

a) 81N, 22W, modelled =shrub tundra, data=Evergreen Taiga/Montane forest – although these biomes are quite different, we note that at a nearby location the modelled biome is also evergreen Taiga

b) 80N, 99W, modelled =temperate deciduous forest, data=Evergreen Taiga/Montane forest.  Again, we see that a nearby location shows the modelled biome to be evergreen Taiga

c) 67N, 141W, modelled =Evergreen Taiga/montaine forest, data =deciduous taiga/montaine forest.  These biomes are both cold forests and have very similar parameters, so there is only a small data-model mismatch.  At a nearby location the data biome is also Evergreen Taiga.

d) 65N, 161W, modelled biome=cool conifer forest, data biome=shrub tundra.  This is a data-model mismatch.  The reason for this data model mismatch is unknown.

Over Europe the biome is not Evergreen Taiga.  Here the sites where the model and data agree are 'warm mixed forest'.  In the text we discuss the model-data mismatch over Europe as:

"Over Western Europe, the warm mixed forest in the model extends too far to the east and the MMM does not reproduce the extent of the cool mixed forests seen in the data.  However, it is quite easy to simulate cool mixed forest in this region with only minor parameter changes to the BIOME4 model (not shown), suggesting model and data are 'close' in this region".

We would like to thank the editor for pointing out that model and data also agree over Eastern Europe.  In the revised version of the manuscript, we write 'Europe' instead of 'Western Europe'. (lines 265-275).

We also correct Eurasia to Asia instead of Eurasia on line 270.

- Line 303: It might be worth also showing the annual mean results in figure 6.

We agree that this could be useful.  However, there is already a lot of information on figure 6, so we  provide the annual mean results as supplementary figure 2.  We also change the text slightly near line 312 to refer to information on this figure.

- Line 484: I think "reason" needs to be changed to "reasons".

Thanks for pointing this out.  This has been corrected.

REPLY TO ARTHUR OLDEMAN

Thank you for reading our paper and providing comments.  We are pleased that you found our study interesting.  We address your comments below:

- L37: the abbreviation SST is not mentioned before, would be good to write out.

Thanks. This has been written out.

- L78: Figure 1: is this annual mean data? It is currently not clarified in the text or caption.

This was an oversight. It is annual mean data and has been clarified (line 84).

- L406-407: 'despite January Arctic SIE being reduced by up to 76%' – where can I find these results? In your manuscript or in a referenced paper? Is this a multi-model mean value?

Thanks for pointing this out. This result is from figure 5 of deNooijer et al (2020). We had cited this in an earlier version of the manuscript, but it has been mistakenly removed. This citation is now included (line 417).

- Table 1: Baatsen et al. is currently out as preprint and can be referred to with doi: https://doi.org/10.5194/cp-2021-140 (at the time of writing, the revised manuscript has been accepted, so you can hopefully refer to the accepted version and doi in your revised manuscript).

Thanks for pointing this out. This has been corrected.

Furthermore, I miss two studies in the introduction and/or discussion that I believe are relevant to this study:

- De Nooijer et al. (2020), that assess Arctic warming in the PlioMIP2 ensemble, https://doi.org/10.5194/cp-16-2325-2020
- Menemenlis et al. (2021, preprint), investigating data-model comparison discrepancies in southwestern US, using a selection of PlioMIP2 models, https://doi.org/10.31223/X5P03R

We agree that De Nooijer should have been cited and will cite this in the introduction of the revised version. This is now cited at line 90. We do not think it is necessary to cite the Menemenlis study, because it looks at precipitation over the southwest US, while our study looks at temperature over the northern high latitudes.

Response to reviewer 2:

We would like to thank the reviewer for reading the paper and for providing useful comments. The reviewers comments are addressed below:

- Despite the title and the focus on "high latitudes", the manuscript does not deal with all high latitude regions. There is a note to the Yallalie data (Perth, Australia) in Figures 1 and 2, but this is located at 30*S and debatable as "high latitude". Otherwise, there is no assessment of southern hemisphere high latitudes. Feng et al. (2017) cited in line 48 was likewise a study of high northern latitudes. The manuscript title and abstract need to clarify this focus.

The reviewer is correct that we did not look at the Pliocene southern high latitudes. We have changed the title to "The Warm Winter Paradox in the Pliocene Northern High Latitudes", and mention 'Northern High latitudes' in the first and second paragraphs of the abstract. We continue to write 'high latitudes' in some other places, where several qualifiers are already used (e.g. line15, which states "Pliocene, high-latitude, terrestrial, summer temperatures").

We also change the Feng reference to refer to the Northern High Latitudes (line 53).

- The abstract is written very clearly, but would benefit from clarifying that the focus of the manuscript is not the whole Pliocene, but a narrower section of this epoch (e.g. lines 28-32 show that the focus is the Mid Pliocene Warm Period, and a shorter interval within it, "KM5c").

We rewrite paragraph 2 of the abstract as:

"We focus on the mid-Pliocene Warm Period (mPWP) and show that understanding the northern high latitude terrestrial temperatures is particularly difficult for the coldest months. Here the temperatures obtained from models and different proxies can vary by more than 20∘C. We refer to this mismatch as the 'warm winter paradox'"

- line 15 might be easier to read with the removal of some commas: "For the Pliocene high latitude terrestrial summer temperatures, models and different proxies are in good agreement".

Thank you for this suggestion. We have removed the commas as advised.

- Introduction, first paragraph: the statements here are logical, but there is no supporting evidence given through citation of literature. Do any of these concerns appear in e.g. the most recent IPCC report, or perhaps discussed in papers looking at data-model (dis)agreement?

In the revised version support these statements through citing:

McClymont et al 2020 and Haywood et al 2016

- Line 29-30 indicates the focus in on the KM5c interval, but lines 75-76 indicate that proxy data for this narrow window is not feasible. This suggests that the model

output is KM5c but the data is MPWP: this needs to be clarified here and in the abstract.

We agree that this is important information, however we think it is too much detail for the abstract.  The information is incorporated here as suggested and we change the paragraph near line 29-30 to:

"This paper will present a DMC for the Pliocene, focussing on the mid-Piacenzian warm period (mPWP, previously referred to as the mid-Pliocene warm period) which occurred between ca 3.3 - 3.0Ma (Dowsett et al., 2016).  Most model simulations represent the KM5C timeslice ($\sim$ 3.205Ma), although data will be less temporally constrained.  The mPWP is the most recent example of a world which had CO2 levels similar to present, and was found by Burke et al. (2018) to be the most similar geological benchmark to global surface temperature predictions of 2030 CE. It is, therefore, a crucial period for model data consensus."

- Line 35: this could refer to subsequent PlioMIP iterations as well as the first one.

This is correct.  We reference PlioMIP2 here as well.

- Paragraph line 40-46: this gives a good overview of the likely uncertainties which were assessed but not a sense of what impact these different uncertainties had (or not). Did any one / combination of the uncertainties give "most likely" cause(s) for some of the mismatches, or was it perhaps uncertainty or location specific? A line of two which notes the main findings of some of these publications would be helpful.

This is a good question.  The uncertainties were different at different locations; hence it is difficult to summarize concisely.  We now write:

"Over land there was greater disagreement between PlioMIP1 models and data than over the oceans, however uncertainties over land are also greater.   Data sites considered by Salzmann et al. (2013) showed uncertainty due to bioclimate range between 0.5degC and 5.8degC, and dating uncertainty of up to 4.0degC.  A modelled range of values was also considered which accounted for variability within the modelled ensemble, CO2 uncertainty and orbital uncertainty.  Including all of these sources of uncertainty allowed models and data to overlap in many places, however the full range of uncertainties was large, meaning it would be difficult to determining the 'true' temperature.  Also, there were still locations where model and data did not agree within the range of the uncertainties. At these locations Salzmann et al. (2013) noted that "the underlying reasons for these large and statistically significant DMC mismatches are unknown".

- Line 73-74: the number of sites able to be directly compared to the models for KM5c is further reduced when a higher threshold for age control was introduced by the Pliovar working group (McClymont et al., 2020, Climate of the Past), which is worth pointing out here as the authors also use that data in Figure 1 and so line

> 80-81 seems to be referring it. I don't think the Pliovar data was used in the PlioMIP2 DMC described in line 73-74.

The reviewer is correct.  The PlioMIP2 DMC described in lines 73-74 did not use the Pliovar data.  We change the text (line 79) to:

"This meant that the 100 ocean sites that were included in a DMC for PlioMIP1 (Dowsett et al., 2013), had reduced to 32 ocean sites for PlioMIP2 (Figure1).

- Line 82-83: "high latitude" here is really "high northern latitude" because there is only a single data point for the entire southern hemisphere, and it has already been described as aligning well with the models (line 82)

In the revised version we change this to high northern latitude as suggested (line 89).

- line 89: the authors could also cite the Pliovar comparison of McClymont (2020) since that also showed better model-data agreement? Aligning with the theme of this manuscript, the Pliovar paper includes a comparison of high northern latitude data and different monthly model outputs, which also indicates a possible role for seasonality in the oceans as well as on land.

As suggested we will  cite  McClymont et al 2020  near line 96 in the revised version.  (Note the software has cut this reference off in the version with tracked changes, but it is seen in the revised version).

- line 169-170: data-model agreement for the PI and "no inherent model bias" – but there seems to be a tendency for the PI MMM (blue dots, Figure 2) to sit either at the low or high end of the PI data (blue dashed lines), which is not noted. Is there a reason for this tendency in the data or models?

We think that the way we have presented this figure might be a bit misleading.  The blue dashed line is the difference between CRU reanalysis data (representing a gridbox) and the point based observations.   We have incorrectly labelled this in the figure as 'PI data uncertainty'.

 Since model results are also gridbox based we would expect the difference between model and point based observation to be similar to that between CRU and point based observation, so the results do suggest "no inherent model bias".

In the revised version of the paper we replace the blue dashed line with a blue triangle.  We also change the labelling in the legend to CRU 1901-1930. We will change the x-axis label to 'Difference between modelled/reanalysis and observed/reconstructed temperatures (degC)".

- line 202: is "prior to 3.5 Ma" late Pliocene as this block is to indicate? How is early/late Pliocene being defined here?

Please note that the response to this comment is different to our original response in the discussion. We have checked the Demske paper and see that the data was prior to 3.5Ma but also after 3.57Ma, which is still Late Pliocene. We have changed the text near line 210 and also in table 3 to reflect this.

We also correct the 'Tnekveem and Hydzhak sites in table 3 which should be labelled as Early Pliocene'

- line 217-218 and some of the preceding sections where DMC is described: I didn't see it defined anywhere, but do the authors consider "agree reasonably well" or to have "similar outputs" between data and models to be the place where there is overlap between the spread of model outputs and the data "points"? There are some places where there is quite a large difference in MMM and data points (e.g. Mirny late Pliocene Figure 3) but where there is overlap in the range of model outputs – does this count as being "similar"?

For the purposes of this paper, it is non-trivial to define quantitatively what is meant by model and data agreeing reasonably well. This is because there are many site-dependant known errors that need accounting for in DMC. We wrote that model and data agreed "reasonably well" because we thought it was useful descriptive language for figure 3; however, what we really wanted to say was that model-data discrepancies for the warm month temperature are very small compared to those for the cold month temperature.

We rewriting lines 217-218 (now ~line 226) as follows:

"Regardless of the exact dating, location or reconstruction method, the DMC over North America follows the same pattern as that seen over North Asia: the modelled temperatures are far too cold for the coldest month, and model-data agreement is much better for the warmest month.

- Line 274: and cite the pliovar DMC?

Agreed we cite Pliovar here (now line 282)

- Figure 6: I became confused here by the KM5c data-model anomalies, because the authors comment that only two sites (the top two) have KM5c data. So, is the KM5c "anomaly" for Mimy and all of the sites beneath this actually "difference between KM5c model and late Pliocene data"? Some revised text in the caption to clarify what these plots represent would help a lot, because at the moment I think it implies that there is KM5c data at all of the sites shown.

Sorry for the confusion. We have rewritten the caption to make it clear which sites are KM5c and which are Late Pliocene.

---

## Author Response (AR2)

At the request of coauthor Ulrich Salzmann we have changed his affiliation to.

Department of Geography and Environmental Sciences , Northumbria University, Newcastle upon Tyne NE1 8ST, UK